# Group Knowledge Transfer:
# Federated Learning of Large CNNs at the Edge

**Chaoyang He    Murali Annavaram    Salman Avestimehr**
University of Southern California
Los Angeles, CA 90007
chaoyang.he@usc.edu   annavara@usc.edu   avestime@usc.edu

## Abstract

Scaling up the convolutional neural network (CNN) size (e.g., width, depth, etc.) is known to effectively improve model accuracy. However, the large model size impedes *training* on resource-constrained edge devices. For instance, federated learning (FL) may place undue burden on the compute capability of edge nodes, even though there is a strong practical need for FL due to its privacy and confidentiality properties. To address the resource-constrained reality of edge devices, we reformulate FL as a group knowledge transfer training algorithm, called FedGKT. FedGKT designs a variant of the alternating minimization approach to train small CNNs on edge nodes and periodically transfer their knowledge by knowledge distillation to a large server-side CNN. FedGKT consolidates several advantages into a single framework: reduced demand for edge computation, lower communication bandwidth for large CNNs, and asynchronous training, all while maintaining model accuracy comparable to FedAvg. We train CNNs designed based on ResNet-56 and ResNet-110 using three distinct datasets (CIFAR-10, CIFAR-100, and CINIC-10) and their non-I.I.D. variants. Our results show that FedGKT can obtain comparable or even slightly higher accuracy than FedAvg. More importantly, FedGKT makes edge training affordable. Compared to the edge training using FedAvg, FedGKT demands 9 to 17 times less computational power (FLOPs) on edge devices and requires 54 to 105 times fewer parameters in the edge CNN. Our source code is released at FedML (https://fedml.ai).

## 1   Introduction

The size of convolutional neural networks (CNN) matters. As seen in both manually designed neural architectures (ResNet [1]) and automated architectures discovered by neural architecture search (DARTS [2], MiLeNAS [3], EfficientNets [4]), scaling up CNN size (e.g., width, depth, etc.) is known to be an effective approach for improving model accuracy. Unfortunately, *training* large CNNs is challenging for resource-constrained edge devices (e.g., smartphones, IoT devices, and edge servers). The demand for edge-based training is increasing as evinced by a recent surge of interest in Federated Learning (FL) [5]. FL is a distributed learning paradigm that can collaboratively train a global model for many edge devices without centralizing any device's dataset [6, 7, 8]. FL can boost model accuracy in situations when a single organization or user does not have sufficient or relevant data. Consequently, many FL services have been deployed commercially. For instance, Google has improved the accuracy of item ranking and language models on Android smartphones by using FL [9]. FL is also a promising solution when data centralization is undesirable or infeasible due to privacy and regulatory constraints [5]. *However*, one significant impediment in edge training is the gap between the computational demand of large CNNs and the meager computational power on edge devices. FL approaches, such as FedAvg [6] can reduce communication frequency by local SGD and model averaging [10], but they only evaluate the convergence property on small CNNs,

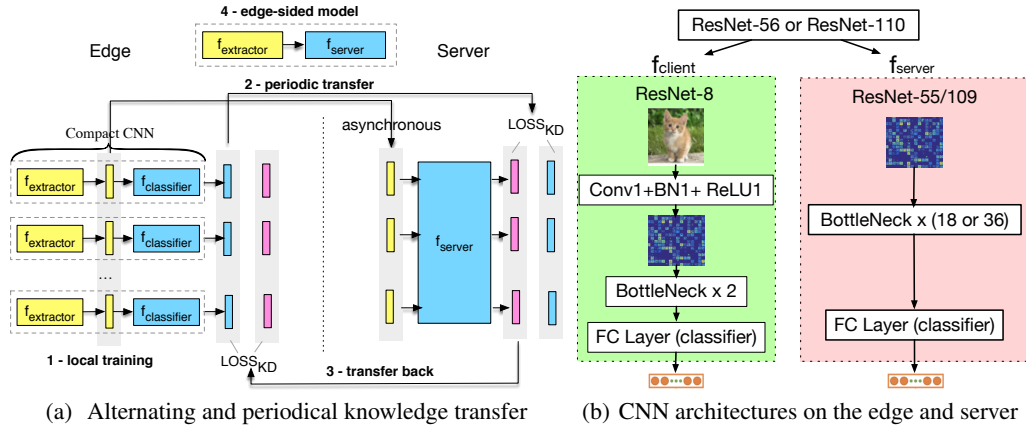

(a) Alternating and periodical knowledge transfer     (b) CNN architectures on the edge and server

Figure 1: Reformulation of Federated Learning: Group Knowledge Transfer

or assume the client has enough computational power with GPUs to train large CNNs, which is unrealistic in a real-world system. To tackle the computational limitation of edge nodes, model parallelism-based split learning (SL) [11, 12] partitions a large model and offloads some portion of the neural architecture to the cloud, but SL has a severe straggler problem because a single mini-batch iteration requires multiple rounds of communication between the server and edges.

In this paper, we propose Group Knowledge Transfer (FedGKT), an efficient federated learning framework for resource-constrained edge devices. FedGKT aims to incorporate benefits from both FedAvg [6] and SL [11, 12] by training using local SGD as in FedAvg but also placing low compute demand at the edge as in SL. FedGKT can transfer knowledge from many compact CNNs trained at the edge to a large CNN trained at a cloud server. The essence of FedGKT is that it reformulates FL as an alternating minimization (AM) approach [13, 14, 15, 16, 17, 18], which optimizes two random variables (the edge model and the server model) by alternatively fixing one and optimizing another. Under this reformulation, FedGKT not only boosts training CNNs at the edge but also contributes to the development of a new knowledge distillation (KD) paradigm, group knowledge transfer, to boost the performance of the server model. Fig. 1(a) provides an overview of FedGKT. The compact CNN on the edge device consists of a lightweight feature extractor and classifier that can be trained efficiently using its private data (*1 - local training*). After local training, all the edge nodes agree to generate *exactly* the same tensor dimensions as an output from the feature extractor. The larger server model is trained by taking features extracted from the edge-side model as inputs to the model, and then uses KD-based loss function that can minimize the gap between the ground truth and soft label (probabilistic prediction in KD [19, 20, 21, 22]) predicted from the edge-side model (*2 - periodic transfer*). To boost the edge model's performance, the server sends its predicted soft labels to the edge, then the edge also trains its local dataset with a KD-based loss function using server-side soft labels (*3 - transfer back*). Thus, the server's performance is essentially boosted by knowledge transferred from the edge models and vice-versa. Once the training is complete, the final model is a combination of its local feature extractor and shared server model (*4 - edge-sided model*). The primary trade-off is that FedGKT shifts the computing burden from edge devices to the powerful server.

FedGKT unifies multiple advantages into a single framework: 1. FedGKT is memory and computation efficient, similar to SL; 2. FedGKT can train in a local SGD manner like FedAvg to reduce the communication frequency; 3. Exchanging hidden features as in SL, as opposed to exchanging the entire model as in FedAvg, reduces the communication bandwidth requirement. 4. FedGKT naturally supports asynchronous training, which circumvents the severe synchronization issue in SL. The server model can immediately start training when it receives inputs from any client. We develop FedGKT based on the FedML research library [23] and comprehensively evaluate FedGKT using edge and server CNNs designed based on ResNet [1] (as shown in Fig. 1(b)). We train on three datasets with varying training difficulties (CIFAR-10 [24], CIFAR-100 [24], and CINIC-10 [25]) and their non-I.I.D. (non identical and independent distribution) variants. As for the model accuracy, our results on both I.I.D. and non-I.I.D. datasets show that FedGKT can obtain accuracy comparable to FedAvg [6]. More importantly, FedGKT makes edge training affordable. Compared to FedAvg, FedGKT demands 9 to 17 times less computational power (FLOPs) on edge devices and requires 54 to 105 times fewer parameters in the edge CNN. To understand FedGKT comprehensively, asynchronous training and ablation studies are performed. Some limitations are also discussed.

## 2   Related Works

**Federated Learning**. Existing FL methods such as FedAvg [6], FedOpt [26], and FedMA [8] face significant hurdles in training large CNNs on resource-constrained devices. Recent works FedNAS [27, 3] and [28] work on large CNNs, but they rely on GPU training to complete the evaluations. Others [29, 30, 31, 32, 33, 34, 35, 36, 37] optimize the communication cost without considering edge computational limitations. Model parallelism-based split learning [11, 12] attempts to break the computational constraint, but it requires frequent communication with the server. **Knowledge Distillation (KD)**. We use KD [19] in a different manner from existing and concurrent works [38, 39, 40, 41, 42, 43, 44, 45]. Previous works only consider transferring knowledge from a large network to a smaller one [19, 20, 21, 22], or they transfer knowledge from a group, but each member in the group shares the same large model architecture or a large portion of the neural architecture with specific tail or head layers [46, 47, 48, 49, 50, 51]. Moreover, all teachers and students in distillation share the same dataset [50, 52, 53, 54], while in our setting each member (client) can only access its own independent dataset. Previous methods use centralized training, but we utilize an alternating training method. **Efficient On-device Deep Learning**. Our work also relates to efficient deep learning on edge devices, such as model compression [55, 56, 57], manually designed architectures (MobileNets [58], ShuffeNets [59], SqueezeNets [60]), or even efficient neural architecture search (EfficientNets [4], FBNet [61]). However, all of these techniques are tailored for the inference phase rather than the training phase.

## 3   Group Knowledge Transfer

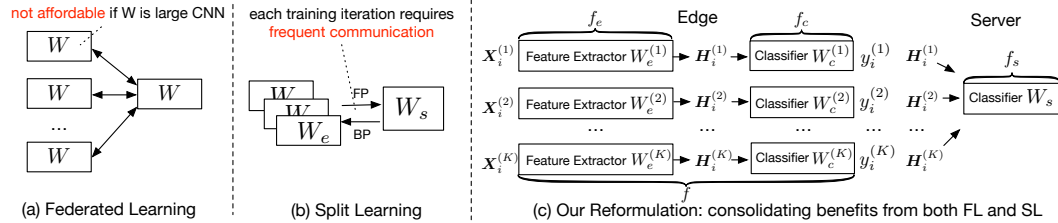

Figure 2: Reformulation of FL: An Alternating Minimization Perspective

### 3.1   Preliminary

We aim to collaboratively train large convolutional neural networks (e.g., ResNet) on many resource-constrained devices that are not equipped with GPU accelerators, without centralizing each device's dataset to the server side. We specifically consider supervised learning with $C$ categories in the entire dataset $\mathcal{D}$. We assume that there are $K$ clients (edge devices) in the network. The $k$th node has its own dataset $\mathcal{D}^k := \left\{ \left( \boldsymbol{X}_i^k, y_i \right) \right\}_{i=1}^{N^{(k)}}$, where $\boldsymbol{X}_i$ is the $i$th training sample, $y_i$ is the corresponding label of $\boldsymbol{X}_i$, $y_i \in \{1, 2, \ldots, C\}$ (a multi-classification learning task), and $N^{(k)}$ is the sample number in dataset $\mathcal{D}^k$. $\mathcal{D} = \{\mathcal{D}_1, \mathcal{D}_2, ..., \mathcal{D}_k\}$, $N = \sum_{k=1}^{K} N^{(k)}$.

In general, we can formulate CNN-based federated learning as a distributed optimization problem:

$$\min_{\boldsymbol{W}} F(\boldsymbol{W}) \overset{\text{def}}{=} \min_{\boldsymbol{W}} \sum_{k=1}^{K} \frac{N^{(k)}}{N} \cdot f^{(k)}(\boldsymbol{W}), \text{where } f^{(k)}(\boldsymbol{W}) = \frac{1}{N^{(k)}} \sum_{i=1}^{N^{(k)}} \ell(\boldsymbol{W}; \boldsymbol{X}_i, y_i) \tag{1}$$

where $\boldsymbol{W}$ represents the network weight of a global CNN in each client. $f^{(k)}(\boldsymbol{W})$ is the $k$th client's local objective function that measures the local empirical risk over the heterogeneous dataset $\mathcal{D}^k$. $\ell$ is the loss function of the global CNN model.

Most off-the-shelf federated optimization methods (e.g., FedAvg [6], FedProx [62], FedNova [63], and FedOpt [26]) propose to solve objective function (1) with variant local SGD [10] optimization methods for communication-efficient training and demonstrate their characteristics with experiments on linear models (logistic regression) or shallow neural networks (2 convolutional layers).

However, as shown in Fig. 2(a), the main drawback is that these methods cannot train large CNN at the *resource-constrained* edge devices due to lack of GPU accelerators and sufficient memory. Model

parallelism-based split learning [11, 12], as shown in Fig. 2(b), attempts to break the computational constraint by splitting $\boldsymbol{W}$ into two portions and offloading the larger portion into the server-side, but a single mini-batch iteration requires remote forward propagation and backpropagation. For edge computing, such a highly frequent synchronization mechanism may lead to the severe straggler problem that significantly slows down the training process.

## 3.2 Reformulation

**Non-convex Optimization**. To solve the resource-constrained problem in existing FL, we reconsider another methodology to solve the FL optimization problem. As illustrated in Fig. 2(c), we divide the global CNN $\boldsymbol{W}$ in Eq. (1) into two partitions: a small feature extractor model $\boldsymbol{W}_e$ and a large-scale server-side model $\boldsymbol{W}_s$, and put them on the edge and the server, respectively. We also add a classifier $\boldsymbol{W}_c$ for $\boldsymbol{W}_e$ to create a small but fully trainable model on the edge. Consequently, we reformulate a single global model optimization into an non-convex optimization problem that requires us to solve the server model $F_s$ and the edge model $F_c$ simultaneously. Our reformulation is as follows:

$$\underset{\boldsymbol{W}_s}{\arg\min} F_s(\boldsymbol{W}_s, \boldsymbol{W}_e^*) = \underset{\boldsymbol{W}_s}{\arg\min} \sum_{k=1}^{K} \sum_{i=1}^{N^{(k)}} \ell_s \left( f_s(\boldsymbol{W}_s; \boldsymbol{H}_i^{(k)}), y_i^{(k)} \right) \tag{2}$$

$$\text{subject to: } \boldsymbol{H}_i^{(k)} = f_e^{(k)}(\boldsymbol{W}_e^{(k)}; \boldsymbol{X}_i^{(k)}) \tag{3}$$

$$\underset{(\boldsymbol{W}_e^{(k)}, \boldsymbol{W}_c^{(k)})}{\arg\min} F_c(\boldsymbol{W}_e^{(k)}, \boldsymbol{W}_c^{(k)}) = \underset{(\boldsymbol{W}_e^{(k)}, \boldsymbol{W}_c^{(k)})}{\arg\min} \sum_{i=1}^{N^{(k)}} \ell_c \left( f^{(k)}((\boldsymbol{W}_e^{(k)}, \boldsymbol{W}_c^{(k)}); \boldsymbol{X}_i^{(k)}), y_i^{(k)} \right) \tag{4}$$

$$= \underset{(\boldsymbol{W}_e^{(k)}, \boldsymbol{W}_c^{(k)})}{\arg\min} \sum_{i=1}^{N^{(k)}} \ell_c \big( f_c^{(k)}(\boldsymbol{W}_c^{(k)}; \underbrace{f_e^{(k)}(\boldsymbol{W}_e^{(k)}; \boldsymbol{X}_i^{(k)})}_{\boldsymbol{H}_i^{(k)}})), y_i^{(k)} \big) \tag{5}$$

Where $\ell_s$ and $\ell_c$ are general loss functions for the server model and the edge model, respectively. $f_s$ is the server model, and $f^{(k)}$ is the edge-side model which consists of feature extractor $f_e^{(k)}$ followed by a classifier $f_c^{(k)}$. $\boldsymbol{W}_s, \boldsymbol{W}_e^{(k)}, \boldsymbol{W}_c^{(k)}$ are the network weights of $f_s, f_e^{(k)}, f_c^{(k)}$, respectively. $\boldsymbol{H}_i^{(k)}$ is $i$-th sample's feature map (a hidden vector or tensor) output by feature extractor $f_e^{(k)}$ (Eq. (3)). Note that Eq. (5) can be solved independently on each client. The $k$th client model $f^{(k)}$ is trained on its local dataset (Eq. (5)), while the server model $f_s$ is trained using $\boldsymbol{H}_i^{(k)}$ as input features (Eq. (2)).

During the inference phase, the final trained model architecture for client $k$ is stacked by the architecture of the feature extractor $f_e^{(k)}$ and the architecture of the server model $f_s$. In practice, the client can either run offline inference by downloading the server model $f_s$ and using it locally or perform online inference through a network connection with the server.

**Advantages and Challenges**. The core advantage of the above reformulation is that when we assume the model size of $f^{(k)}$ is multiple orders of magnitude smaller than that of $f_s$, the edge training is affordable. Moreover, as discussed in [11, 12], for large CNN training, the communication bandwidth for transferring $\boldsymbol{H}_i^{(k)}$ to the server is substantially less than communicating all model parameters as is done in traditional federated learning.

Conversely, we also observe the difficulty of the reformulated optimization problem. First, each client is expected to adequately solve the inner optimization (Eq. (5)). Namely, each client should train its feature extractor $f_e^{(k)}$ well to ensure that Eq. (3) can accurately generate $\boldsymbol{H}_i^{(k)}$ for any given input image. However, in the FL setting, the dataset on each edge device is small and thus may be inadequate in training a CNN-based feature extractor solely based on the local dataset. In addition, the outer optimization Eq. (2) and inter optimization Eq. (5) are correlated: Eq. (2) relies on the quality of $\boldsymbol{H}_i^{(k)}$ which is optimized by Eq. (5). This correlation further makes the outer optimization Eq. (2) difficult to converge if the individual client-side feature extractors $f_e^{(k)}$ are not trained adequately.

## 3.3 Group Knowledge Transfer (FedGKT)

**Scaling Edge Dataset Limitations with Knowledge Transfer**. Given the above challenges, we incorporate knowledge distillation loss into the optimization equations to circumvent the optimization

difficulty. The intuition is that knowledge transferred from the the server model can boost the optimization on the edge (Eq. (5)). As such, we propose to transfer group knowledge bidirectionally. The server CNN absorbs the knowledge from many edges, and an individual edge CNN obtains enhanced knowledge from the server CNN. To be more specific, in Eq. (2) and (5), we design $\ell_s$ and $\ell_c$ as follows.

$$\ell_s = \ell_{CE} + \sum_{k=1}^{K} \ell_{KD}\left(\boldsymbol{z}_s, \boldsymbol{z}_c^{(k)}\right) = \ell_{CE} + \sum_{k=1}^{K} D_{KL}\left(\boldsymbol{p}_k \| \boldsymbol{p}_s\right) \qquad (6)$$

$$\ell_c^{(k)} = \ell_{CE} + \ell_{KD}\left(\boldsymbol{z}_s, \boldsymbol{z}_c^{(k)}\right) = \ell_{CE} + D_{KL}\left(\boldsymbol{p}_s \| \boldsymbol{p}_k\right) \qquad (7)$$

$\ell_{CE}$ is the cross-entropy loss between the predicted values and the ground truth labels. $D_{KL}$ is the Kullback Leibler (KL) Divergence function that serves as a term in the loss function $\ell_s$ and $\ell_c$ to transfer knowledge from a network to another. $\boldsymbol{p}_k^i = \frac{\exp\left(z_c^{(k,i)}/T\right)}{\sum_{i=1}^{C}\exp\left(z_c^{(k,i)}/T\right)}$ and $\boldsymbol{p}_s^i = \frac{\exp\left(z_s^i/T\right)}{\sum_{i=1}^{C}\exp\left(z_s^i/T\right)}$.

They are the probabilistic prediction of the edge model $f^{(k)}$ and the server model $f_s$, respectively. They are calculated with the softmax of logits $\boldsymbol{z}$. The logit $\boldsymbol{z}_s$ and $\boldsymbol{z}_c^{(k)}$ are the output of the last fully connected layer in the server model and the client model, respectively. T is the temperature hyperparameter of the softmax function.

Intuitively, the KL divergence loss attempts to bring the soft label and the ground truth close to each other. In doing so, the server model absorbs the knowledge gained from each of the edge models. Similarly, the edge models attempt to bring their predictions closer to the server model's prediction and thereby absorb the server model knowledge to improve their feature extraction capability.

**Improved Alternating Minimization**. After plugging Eq. (6) and (7) into our reformulation (Eq. (2) and (5)), we propose a variant of Alternating Minimization (AM) [13, 14, 15, 16, 17, 18] to solve the reformulated optimization problem as follows:

$$\underset{\boldsymbol{W}_s}{\operatorname{argmin}} F_s(\boldsymbol{W}_s, \boldsymbol{W}_e^{(k)*}) = \underset{\boldsymbol{W}_s}{\operatorname{argmin}} \sum_{k=1}^{K} \sum_{i=1}^{N^{(k)}} \ell_{CE}\big(f_s(\boldsymbol{W}_s; \underbrace{f_e^{(k)}(\boldsymbol{W}_e^{(k)*}; \boldsymbol{X}_i^{(k)})}_{\boldsymbol{H}_i^{(k)}}), y_i^{(k)}\big) + \sum_{k=1}^{K} \ell_{KD}\big(\boldsymbol{z}_c^{(k)*}, \boldsymbol{z}_s\big)$$
$$\qquad (8)$$

$$where \ \ \boldsymbol{z}_c^{(k)*} = f_c^{(k)}(\boldsymbol{W}_c^{(k)}; \underbrace{f_e^{(k)}(\boldsymbol{W}_e^{(k)*}; \boldsymbol{X}_i^{(k)})}_{\boldsymbol{H}_i^{(k)}})), and \ \boldsymbol{z}_s = f_s(\boldsymbol{W}_s; \boldsymbol{H}_i^{(k)}) \qquad (9)$$

$$\underset{\boldsymbol{W}^{(k)}}{\operatorname{argmin}} F_c(\boldsymbol{W}_s^*, \boldsymbol{W}^{(k)}) = \underset{\boldsymbol{W}^{(k)}}{\operatorname{argmin}} \sum_{i=1}^{N^{(k)}} \ell_{CE}\big(f_c^{(k)}(\boldsymbol{W}_c^{(k)}; \underbrace{f_e^{(k)}(\boldsymbol{W}_e^{(k)}; \boldsymbol{X}_i^{(k)})}_{\boldsymbol{H}_i^{(k)}})), y_i^{(k)}\big) + \ell_{KD}\big(\boldsymbol{z}_s^*, \boldsymbol{z}_c^{(k)}\big)$$
$$\qquad (10)$$

$$where \ \ \boldsymbol{z}_c^{(k)} = f_c^{(k)}(\boldsymbol{W}_c^{(k)}; \underbrace{f_e^{(k)}(\boldsymbol{W}_e^{(k)}; \boldsymbol{X}_i^{(k)})}_{\boldsymbol{H}_i^{(k)}})), and \ \boldsymbol{z}_s^* = f_s(\boldsymbol{W}_s^*; \boldsymbol{H}_i^{(k)}) \qquad (11)$$

Where the $*$ superscript notation in above equations presents related random variables are fixed during optimization. $\boldsymbol{W}^{(k)}$ is the combination of $\boldsymbol{W}_e^{(k)}$ and $\boldsymbol{W}_c^{(k)}$. AM is a solver in convex and non-convex optimization theory and practice that optimizes two random variables alternatively. In Eq. (8), we fix $\boldsymbol{W}^{(k)}$ and optimize (train) $\boldsymbol{W}_s$ for several epochs, and then we switch to (10) to fix $\boldsymbol{W}_s$ and optimize $\boldsymbol{W}^{(k)}$ for several epochs. This optimization occurs throughout many rounds between Eq. (8) and (10) until reaching a convergence state.

**Key Insight**. The essence of our reformulation is that the alternating minimization (Eq. (8) and Eq. (10)) uses knowledge distillation across all edges to simplify the optimization, which scales the dataset limitation on each edge in federated learning. In particular, we achieve this objective using a local cross-entropy loss computed based only on the ground truth and the model output, and a second loss that uses the KL divergence across edges and the server, which effectively captures the contribution (knowledge) from multiple client datasets. Moreover, each minimization subproblem can be solved with SGD and its variants (e.g., SGD with momentum [64], ADAM [65, 66]).

**Algorithm 1 Group Knowledge Transfer**. The subscript $s$ and $k$ stands for the server and the $k$th edge, respectively. $E$ is the number of *local* epochs, $T$ is the number of communication rounds; $\eta$ is the learning rate; $\boldsymbol{X}^{(k)}$ represents input images at edge $k$; $\boldsymbol{H}^{(k)}$ is the extracted feature map from $\boldsymbol{X}^{(k)}$; $\boldsymbol{Z}_s$ and $\boldsymbol{Z}_c^{(k)}$ are the logit tensor from the client and the server, respectively.

1: **ServerExecute()**:
2: **for** each round $t = 1, 2, ..., T$ **do**
3:     **for** each client $k$ **in parallel do**
4:         // the server broadcasts $\boldsymbol{Z}_c^{(k)}$ to the client
5:         $\boldsymbol{H}^{(k)}, \boldsymbol{Z}_c^{(k)}, \boldsymbol{Y}^{(k)} \leftarrow$ **ClientTrain**$(k, \boldsymbol{Z}_s^{(k)})$
6:     $\boldsymbol{Z}_s \leftarrow$ empty dictionary
7:     **for** each local epoch $i$ from 1 to $E_s$ **do**
8:         **for** each client $k$ **do**
9:             **for** $idx, \boldsymbol{b} \in \{\boldsymbol{H}^{(k)}, \boldsymbol{Z}_c^{(k)}, \boldsymbol{Y}^{(k)}\}$ **do**
10:                $\boldsymbol{W}_s \leftarrow \boldsymbol{W}_s - \eta_s \nabla \ell_s(\boldsymbol{W}_s; \boldsymbol{b})$
11:                **if** $i == E_s$ **then**
12:                    $\boldsymbol{Z}_s^{(k)}[idx] \leftarrow f_s(\boldsymbol{W}_s; \boldsymbol{h}^{(k)})$
13:     // illustrated as "transfer back" in Fig. 1(a)
14:     **for** each client $k$ **in parallel do**
15:         send the server logits $\boldsymbol{Z}_s^{(k)}$ to client $k$
16:

17: **ClientTrain**$(k, \boldsymbol{Z}_s^{(k)})$:
18: // illustrated as "local training "in Fig. 1(a)
19: **for** each local epoch $i$ from 1 to $E_c$ **do**
20:     **for** batch $\boldsymbol{b} \in \{\boldsymbol{X}^{(k)}, \boldsymbol{Z}_s^{(k)}, \boldsymbol{Y}^{(k)}\}$ **do**
21:         // $\ell_c^{(k)}$ is computed using Eq. (7)
22:         $\boldsymbol{W}^{(k)} \leftarrow \boldsymbol{W}^{(k)} - \eta_k \nabla \ell_c^{(k)}(\boldsymbol{W}^{(k)}; \boldsymbol{b})$
23: // extract features and logits
24: $\boldsymbol{H}^{(k)}, \boldsymbol{Z}_c^{(k)} \leftarrow$ empty dictionary
25: **for** $idx$, batch $\boldsymbol{x}^{(k)}, \boldsymbol{y}^{(k)} \in \{\boldsymbol{X}^{(k)}, \boldsymbol{Y}^{(k)}\}$ **do**
26:     $\boldsymbol{h}^{(k)} \leftarrow f_e^{(k)}(\boldsymbol{W}_e^{(k)}; \boldsymbol{x}^{(k)})$
27:     $\boldsymbol{z}_c^{(k)} \leftarrow f_c(\boldsymbol{W}_c^{(k)}; \boldsymbol{h}^{(k)})$
28:     $\boldsymbol{H}^{(k)}[idx] \leftarrow \boldsymbol{h}^{(k)}$
29:     $\boldsymbol{Z}_c^{(k)}[idx] \leftarrow \boldsymbol{z}_c^{(k)}$
30: return $\boldsymbol{H}^{(k)}, \boldsymbol{Z}_c^{(k)}, \boldsymbol{Y}^{(k)}$ to server

**Training Algorithm.** To elaborate, we illustrate the alternating training algorithm FedGKT in Fig. 1(a) and summarize it as Algorithm 1. During each round of training, the client uses local SGD to train several epochs and then sends the extracted feature maps and related logits to the server. When the server receives extracted features and logits from each client, it trains the much larger server-side CNN. The server then sends back its global logits to each client. This process iterates over multiple rounds, and during each round the knowledge of all clients is transferred to the server model and vice-versa. For the FedGKT training framework, the remaining step is to design specific neural architectures for the client model and the server model. To evaluate the effectiveness of FedGKT, we design CNN architectures based on ResNet [1], which are shown in Fig. 1(b). More details can also be found in Appendix B.3.

## 4 Experiments

### 4.1 Experimental Setup

**Implementation**. We develop the FedGKT training framework based on `FedML` [23], an open source federated learning research library that simplifies the new algorithm development and deploys it in a distributed computing environment. Our server node has 4 NVIDIA RTX 2080Ti GPUs with sufficient GPU memory for large model training. We use several CPU-based nodes as clients training small CNNs.

**Task and Dataset**. Our training task is image classification on CIFAR-10 [24], CIFAR-100 [24], and CINIC-10 [25]. We also generate their non-I.I.D. variants by splitting training samples into $K$ unbalanced partitions. Details of these three datasets are introduced in Appendix A.1. The test images are used for a global test after each round. For different methods, we record the top 1 test accuracy as the metric to compare model performance. Note that we do not use LEAF [67] benchmark datasets because the benchmark models provided are tiny models (CNN with only two convolutional layers) or the datasets they contain are too easy for modern CNNs (e.g., Federated EMNIST), which are unable to adequately evaluate our algorithm running on large CNN models. Compared to LEAF, FedML [23] benchmark supports CIFAR-10, CIFAR-100, and CINIC-10 (contains images from ImageNet).

**Baselines**. We compare FedGKT with state-of-the-art FL method FedAvg [6], and a centralized training approach. Split Learning-based method [11, 12] is used to compare the communication cost. Note that we do not compare with FedProx [62] because it performs worse than FedAvg in the large CNN setting, as demonstrated in [8]. We also do not compare with FedMA [8] because it cannot work on modern DNNs that contain batch normalization layers (e.g., ResNet).

**Model Architectures**. Two modern CNN architectures are evaluated: ResNet-56 and ResNet-110 [1]. The baseline FedAvg requires all edge nodes to train using these two CNNs. For FedGKT, the edge and server-sided models are designed based on these two CNNs. On the edges, we design a tiny CNN architecture called ResNet-8, which is a compact CNN containing 8 convolutional layers (described in Fig. 1(b) and Table 7 in Appendix). The server-sided model architectures are ResNet-55 and ResNet-109 (Table 8 and 9 in Appendix), which have the same input dimension to match the output of the edge-sided feature extractor. For split learning, we use the extractor in ResNet-8 as the edge-sided partition of CNNs, while the server-side partitions of CNN are also ResNet-55 and ResNet-109.

## 4.2 Result of Model Accuracy

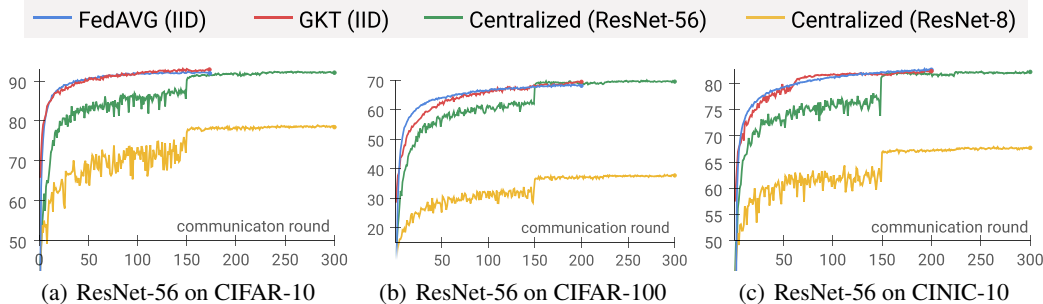

(a) ResNet-56 on CIFAR-10     (b) ResNet-56 on CIFAR-100     (c) ResNet-56 on CINIC-10

Figure 3: The Test Accuracy of ResNet-56 (Edge Number = 16)

For standard experiments, we run on 16 clients and a GPU server for all datasets and models. Fig. 3 shows the curve of the test accuracy during training on ResNet-56 model with 3 datasets. It includes the result of centralized training, FedAvg, and FedGKT. We also summarize all numerical results of ResNet-56 and ResNet-110 in Table 1. In both I.I.D. and non-I.I.D. setting, FedGKT obtains comparable or even better accuracy than FedAvg.

**Hyperparameters**. There are four important hyper-parameters in our FedGKT framework: the communication round, as stated in line #2 of Algorithm 1, the edge-side epoch number, the server-side epoch number, and the server-side learning rate. After a tuning effort, we find that the edge-side epoch number can simply be 1. The server epoch number depends on the data distribution. For I.I.D. data, the value is 20, and for non-I.I.D., the value depends on the level of data bias. For I.I.D., Adam optimizer [65] works better than SGD with momentum [64], while for non-I.I.D., SGD with momentum works better. During training, we reduce the learning rate once the accuracy has plateaued [68, 69]. We use the same data augmentation techniques for fair comparison (random crop, random horizontal flip, and normalization). More details of hyper-parameters are described in Appendix B.4.

Table 1: The Test Accuracy of ResNet-56 and ResNet-110 on Three Datasets.

| Model | Methods | CIFAR-10 | | CIFAR-100 | | CINIC-10 | |
|---|---|---|---|---|---|---|---|
| | | I.I.D. | non-I.I.D. | I.I.D. | non-I.I.D. | I.I.D. | non-I.I.D. |
| ResNet-56 | **FedGKT (ResNet-8, ours)** | **92.97** | **86.59** | **69.57** | **63.76** | **81.51** | **77.80** |
| | FedAvg (ResNet-56) | 92.88 | 86.60 | 68.09 | 63.78 | 81.62 | 77.85 |
| | Centralized (ResNet-56) | 93.05 | | 69.73 | | 81.66 | |
| | Centralized (ResNet-8) | 78.94 | | 37.67 | | 67.72 | |
| ResNet-110 | **FedGKT (ResNet-8, ours)** | **93.47** | **87.18** | **69.87** | **64.31** | **81.98** | **78.39** |
| | FedAvg (ResNet-110) | 93.49 | 87.20 | 68.58 | 64.35 | 82.10 | 78.43 |
| | Centralized (ResNet-110) | 93.58 | | 70.18 | | 82.16 | |
| | Centralized (ResNet-8) | 78.94 | | 37.67 | | 67.72 | |

*Note: 1. It is a normal phenomenon when the test accuracy in non-I.I.D. is lower than that of I.I.D.. This is confirmed by both this study and other CNN-based FL works [28, 26]; 2. In the non-I.I.D. setting, since the model performance is sensitive to the data distribution, we fix the distribution of non-I.I.D. dataset for a fair comparison. Appendix A.2 describes the specific non-I.I.D. distribution used in the experiment; 3. Table 10,11,12 in Appendix summarize the corresponding hyperparameters used in the experiments.

## 4.3 Efficiency Evaluation

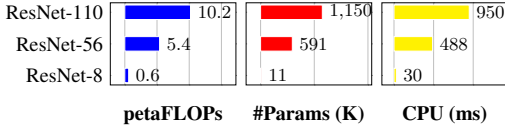

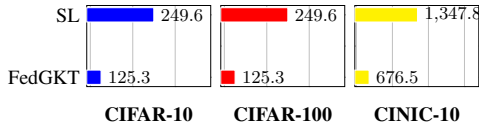

Figure 4: Edge Computational Efficiency (CIFAR-100)

Figure 5: Communication Efficiency (ResNet-56)

To compare the computational demand on training, we count the number of FLOPs (floating-point operations) performed on edge using prior methods [70, 71]. We report the result on CIFAR-100 in Fig. 4. Compared to the FedAvg baseline, the computational cost on the edge of our FedGKT (ResNet-8) is 9 times less than that of ResNet-56 and 17 times less than that of ResNet-110 (The memory cost comparison can be roughly compared by the model parameter number: ResNet-8 has 11K parameters, which is 54 times less than that of ResNet-56 and 105 times less than that of ResNet-110. We also test the CPU running time per mini-batch (batch size is 64) forward-backward propagation on Intel i7 CPU (which has a more aggressive performance than current edge devices). The results show that ResNet-8 requires only 3% of ResNet-110's training time (30 ms v.s. 950 ms).

To compare communication costs, we use SL [11, 12] as the baseline, which also exchanges hidden feature maps rather than the entire model. The communication cost is calculated using Eq. (12) and (13) in Appendix B.2 without using data compression techniques. The results are shown in Fig. 5 (X-axis units: GBytes). FedGKT uses fewer feature map exchanges with the server than SL.

## 4.4 Ablation Study: Understanding FedGKT under Different Settings

Table 2: Ablation Study on Loss Functions

|  | CIFAR-10 | CIFAR-100 | CINIC-10 |
|---|---|---|---|
| None | -/diverge | -/diverge | -/diverge |
| S–>E | 92.97 | 68.44 | 81.51 |
| S<–>E | 90.53 | 69.57 | 80.01 |

Table 3: Asynchronous Training

|  | CIFAR-10 | CIFAR-100 | CINIC-10 |
|---|---|---|---|
| Sync | 92.97 | 69.57 | 81.51 |
| Async | 92.92 | 69.65 | 81.43 |

**The Effectiveness of Knowledge Transfer**. Table 2 shows the results on the efficacy of using distillation loss $\ell_{KD}$ in Eq. (7) and Eq. (6). We created a scenario in which both the client and server only use $\ell_{CE}$ without using $\ell_{KD}$ (labeled *None*). In this setting, the accuracy is low (e.g., 40%) or the training diverges (uniformly notated as "-/diverge"). In another scenario, only the clients use $\ell_{KD}$ to update their local models, but the server does not (noted as single directional transfer *S->E*). We observe that the transfer from the server to the edge is always helpful, while the bidirectional transfer (*S<–>E*) is more effective as the dataset becomes increasingly difficult (CIFAR-100).

**Asynchronous Training**. Since the server does not need to wait for updates from all clients to start training, FedGKT naturally supports asynchronous training. We present the experimental results in Table 3. The result shows that asynchronous training does not negatively affect model accuracy. This demonstrates the advantage of our method over SL, in which every edge requires multiple synchronizations for each mini-batch iteration.

Table 4: FedGKT with Different # of Edge

|  | 8 | 16 | 64 | 128 |
|---|---|---|---|---|
| FedGKT | 69.51 | 69.57 | 69.65 | 69.59 |

Table 5: Small CNNs on CIFAR-10

|  | ResNet-4 | ResNet-6 | ResNet-8 |
|---|---|---|---|
| Test Accuracy | 88.86 | 90.32 | 92.97 |

**FedGKT with Different Edge Number.** To understand the scalability of FedGKT, we evaluate its performance with varying edge nodes. The test accuracy results are shown in Table 4. In general, adding more edge nodes does not negatively affect accuracy.

**Smaller Architectures.** We test the performance of FedGKT using even smaller edge models: ResNet-4 and ResNet-6 on CIFAR-10. ResNet-4 and ResNet-6 use one and two BasicBlock components (including two convolutional layers), respectively. The result is shown in Table 5. While reducing the edge model size to ResNet-8 did not reduce accuracy, when the model size is reduced even more substantially, it does reduce the overall accuracy.

## 5 Discussion

Federated learning (FL) is an art of trade-offs among many aspects, including model accuracy, data privacy, computational efficiency, communication cost, and scalability. We recognize the challenges of developing a universal method that can address all problems; thus, we discuss some limitations of our method.

**1. Privacy and robustness:** [72] shows we can backdoor federated learning. Although our work does not address the privacy concern, we believe existing methods such as differential privacy (DP) and multi-party computation (MPC) can defend the data privacy from the hidden vector reconstruction attack. Intuitively, exchanging hidden feature maps is safer than exchanging the model or gradient. Note that the hidden map exchange happens at the training phase. This consequently makes the attack more difficult because the attacker's access is limited to the evolving and untrained feature map rather than the fully trained feature map that represents the raw data. Given that the model and gradient exchange may also leak privacy, the lack of analysis and comparison of the degree of privacy leakages between these three settings (gradient, model, and hidden map) is the first limitation of our work.

**2. Communication cost:** compared to the entire model weight or gradient, the hidden vector is definitely much smaller (e.g., the hidden vector size of ResNet-110 is around 64KB while the entire gradient/model size is 4.6MB for 32x32 images). Even in the high resolution vision tasks settings, this observation also holds (e.g., when image size is 224x224, the hidden feature map size is only 1Mb, compared to the size of ResNet 100Mb). Since the hidden vector for each data point can be transmitted independently, FedGKT has a smaller bandwidth requirement than gradient or model exchange. However, our proposed method has a potential drawback in that the total communication cost depends on the number of data points, although our experimental results demonstrate that our method has smaller communication costs than split learning because of fewer communication rounds for convergence. In settings where the sample number is extremely large and the image resolution is extremely high, both our method and split learning would have a high communication cost in total.

**3. Label deficiency:** The proposed FedGKT can only work on supervised learning. However, label deficiency is a practical problem that cannot be ignored. Many application cases do not have sufficient labels, since it is difficult to design mechanisms to incentivize users to label their private local data.

**4. Scalability (a large number of clients):** in the cross-device setting, we need to collaboratively train models with numerous smartphones (e.g., if the client number is as high as 1 million). One way to mitigate the scalability is by selecting clients in each round with a uniform sampling strategy [6]. We run experiments under this setting but found that this sampling method requires many more rounds of training to converge. Even though the communication cost is acceptable, this sampling method is still imperfect in practice ([9] describes many constraints that a production system might face). We argue that uniform sampling may not be the best practice and that scalability is a common limitation for most existing works. In summary, we concede that our proposed method does not have an advantage in addressing the scalability challenge.

**5. Model personalization:** the final trained model under our FedGKT framework is a combination of the global server model and the client model, which is a potential method to help clients learn personalized models. For example, we can fine-tune the client model for several epochs to see if the combination of such a personalized client model and the server model is more effective. We do not explicitly demonstrate this in our experiments, but we hope to explore this possibility in future works.

## 6 Conclusion

In this work, to tackle the resource-constrained reality, we reformulate FL as a group knowledge transfer (FedGKT) training algorithm. FedGKT can efficiently train small CNNs on edges and periodically transfer their knowledge by knowledge distillation to a server-side CNN with a large capacity. FedGKT achieves several advantages in a single framework: reduced demand for edge computation, lower communication cost for large CNNs, and asynchronous training, all while maintaining model accuracy comparable to FL. To simplify the edge training, we also develop a distributed training system based on our FedGKT. We evaluate FedGKT by training modern CNN architectures (ResNet-56 and ResNet-110) on three distinct datasets (CIFAR-10, CIFAR-100, and CINIC-10) and their non-I.I.D. variants. Our results show that FedGKT can obtain comparable or even slightly higher accuracy. More importantly, FedGKT makes edge training affordable. Compared to the edge training using FedAvg, FedGKT costs 9 to 17 times less computational power (FLOPs) and requires 54 to 105 times fewer parameters.

## Broader Impact

FedGKT can efficiently train large deep neural networks (CNNs) in resource-constrained edge devices (such as smartphones, IoT devices, and edge servers). Unlike past FL approaches, FedGKT demonstrates the feasibility of training a large server-side model by using many small client models. FedGKT preserves the data privacy requirements of the FL approach but also works within the constraints of an edge computing environment. Smartphone users may benefit from this technique because their private data is protected, and they may also simultaneously obtain a high-quality model service. Organizations such as hospitals, and other non-profit entities with limited training resources, can collaboratively train a large CNN model without revealing their datasets while achieving significant training cost savings. They can also meet requirements regarding the protection of intellectual property, confidentiality, regulatory restrictions, and legal constraints.

As for the potential risks of our method, a client can maliciously send incorrect hidden feature maps and soft labels to the server, which may potentially impact the overall model accuracy. These effects must be detected and addressed to maintain overall system stability. Second, the relative benefits for each client may vary. For instance, in terms of fairness, edge nodes which have smaller datasets may obtain more model accuracy improvement from collaborative training than those which have a larger amount of training data. Our training framework does not consider how to balance this interest of different parties.

## Acknowledgments

This material is based upon work supported by Defense Advanced Research Projects Agency (DARPA) under Contract No. HR001117C0053 and FA8750-19-2-1005, ARO award W911NF1810400, NSF grants CCF-1703575 and CCF-1763673, and ONR Award No. N00014-16-1-2189. The views, opinions, and/or findings expressed are those of the author(s) and should not be interpreted as representing the official views or policies of the Department of Defense or the U.S. Government.

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
