[Supplementary Material]

# A Datasets

## A.1 A Summary of Dataset Used in Experiments

**CIFAR-10** [24] consists of 60000 32×32 colour images in 10 classes, with 6000 images per class. There are 50000 training images and 10000 test images. **CIFAR-100** [24] has the same amount of samples as CIFAR-10, but it is a more challenging dataset since it has 100 classes containing 600 images each. **CINIC-10** [25] has a total of 270,000 images, 4.5 times that of CIFAR-10. It is constructed from two different sources: ImageNet and CIFAR-10. It is not guaranteed that the constituent elements are drawn from the same distribution. This characteristic fits for federated learning because we can evaluate how well models cope with samples drawn from similar but not identical distributions. CINIC-10 has three sub-datasets: training, validation, and testing. We train on the training dataset and test on the testing, without using the validation dataset for all experiments. Our source code provides the link to download these three datasets.

For the non-I.I.D. dataset, the partition is unbalanced: sampling $\mathbf{p}_c \sim \text{Dir}_J(0.5)$ and allocating a $\mathbf{p}_{c,k}$ proportion of the training samples of class $c$ to local client $k$.

## A.2 Heterogeneous Distribution (non-I.I.D.) in Each Client

We fix the non-I.I.D. distribution to fairly compare different methods. Table 6 is a specific distribution used in the experiments. We also conduct experiments in other non-I.I.D. distributions and observe that our FedGKT method also outperforms baselines. To generate the different distribution, we can change the random seed in *main.py* of our source code.

| Client ID | Numbers of Samples in the Classes | | | | | | | | | | Distribution |
|---|---|---|---|---|---|---|---|---|---|---|---|
| | $c_0$ | $c_1$ | $c_2$ | $c_3$ | $c_4$ | $c_5$ | $c_6$ | $c_7$ | $c_8$ | $c_9$ | |
| k=0 | 144 | 94 | **1561** | 133 | **1099** | 1466 | **0** | **0** | **0** | **0** | |
| k=1 | 327 | 28 | 264 | 16 | 354 | 2 | 100 | 20 | 200 | 3 | |
| k=2 | 6 | 6 | 641 | 1 | 255 | 4 | 1 | 2 | 106 | **1723** | |
| k=3 | 176 | 792 | 100 | 28 | 76 | 508 | 991 | 416 | 215 | 0 | |
| k=4 | 84 | **1926** | 1 | 408 | 133 | 24 | 771 | **0** | **0** | **0** | |
| k=5 | 41 | 46 | 377 | 541 | 7 | 235 | 54 | **1687** | 666 | **0** | |
| k=6 | 134 | 181 | 505 | 720 | 123 | 210 | 44 | 58 | 663 | 221 | |
| k=7 | 87 | 2 | 131 | **1325** | **1117** | 704 | **0** | **0** | **0** | **0** | |
| k=8 | 178 | 101 | 5 | 32 | **1553** | 10 | 163 | 9 | 437 | 131 | |
| k=9 | 94 | 125 | **0** | 147 | 287 | 100 | 23 | 217 | 608 | 279 | |
| k=10 | 379 | 649 | 106 | 90 | 35 | 119 | 807 | 819 | 3 | 85 | |
| k=11 | **1306** | 55 | 681 | 227 | 202 | 34 | **0** | 648 | **0** | **0** | |
| k=12 | **1045** | 13 | 53 | 6 | 77 | 70 | 482 | 7 | 761 | 494 | |
| k=13 | 731 | 883 | 15 | 161 | 387 | 552 | 4 | **1051** | **0** | **0** | |
| k=14 | 4 | 97 | 467 | 899 | **0** | 407 | 50 | 64 | **1098** | 797 | |
| k=15 | 264 | 2 | 93 | 266 | 412 | 142 | 806 | 2 | 243 | **1267** | |

Table 6: The actual heterogeneous data distribution (non-I.I.D.) generated from CIFAR-10

# B Extra Experimental Results and Details

## B.1 Computational Efficiency on CIFAR-10 and CINIC-10

Figure 6: Edge Computational Efficiency (CIFAR-100)    Figure 7: Edge Computational Efficiency (CINIC-10)

## B.2 The Method of Communication Cost Calculation

For split learning (SL), the method to calculate the communication cost is:

$$\textit{Communication Cost of SL} = (\textit{the size of the hidden feature map} + \textit{the size of the gradient in the split layer})$$
$$\times(\textit{number of samples in dataset}) \times (\textit{number of epochs}) \tag{12}$$

For FedGKT, the method to calculate the communication cost is:

$$Communication\ Cost\ of\ FedGKT = (\textit{the size of the hidden feature map}+$$
$$\textit{the size of soft labels received from the server side}) \times \textit{(number of samples in dataset)}$$
$$\times\textit{(number of communication rounds)} \quad (13)$$

## B.3 Details of Convolutional Neural Architecture on Edge and Server

ResNet-8 is a compact CNN. Its head convolutional layer (including batch normalization and ReLU non-linear activation) is used as the feature extractor. The remaining two Bottlenecks (a classical component in ResNet, each containing 3 convolutional layers) and the last fully-connected layer are used as the classifier.

Table 7: Detailed information of the ResNet-8 architecture used in our experiment

| Layer | Parameter & Shape (cin, cout, kernal size) & hyper-parameters | # |
|---|---|---|
| | conv1: $3 \times 16 \times 3 \times 3$, stride:(1, 1); padding:(1, 1) | $\times 1$ |
| | maxpool: $3 \times 1$ | $\times 1$ |
| layer1 | conv1: $16 \times 16 \times 3 \times 3$, stride:(1, 1); padding:(1, 1) <br> conv2: $16 \times 16 \times 3 \times 3$, stride:(1, 1); padding:(1, 1) | $\times 2$ |
| | avgpool | $\times 1$ |
| | fc: $16 \times 10$ | $\times 1$ |

Table 8: Detailed information of the ResNet-55 architecture used in our experiment

| Layer | Parameter & Shape (cin, cout, kernel size) & hyper-parameters | # |
|---|---|---|
| layer1 | conv1: $16 \times 16 \times 1 \times 1$, stride:(1, 1) <br> conv2: $16 \times 16 \times 3 \times 3$, stride:(1, 1); padding:(1, 1) <br> conv3: $16 \times 64 \times 1 \times 1$, stride:(1, 1) <br> downsample.conv: $16 \times 64 \times 1 \times 1$, stride:(1, 1) | $\times 1$ |
| | conv1: $64 \times 16 \times 1 \times 1$, stride:(1,1) <br> conv2: $16 \times 16 \times 3 \times 3$, stride:(1, 1), padding:(1,1) <br> conv3: $16 \times 64 \times 1 \times 1$, stride:(1, 1) | $\times 5$ |
| layer2 | conv1: $64 \times 32 \times 1 \times 1$, stride:(1, 1) <br> conv2: $32 \times 32 \times 3 \times 3$, stride:(2, 2); padding:(1, 1) <br> conv3: $32 \times 128 \times 1 \times 1$, stride:(1, 1) <br> downsample.conv: $64 \times 128 \times 1 \times 1$, stride:(2, 2) | $\times 1$ |
| | conv1: $128 \times 32 \times 1 \times 1$, stride:(1, 1)] <br> conv2: $32 \times 32 \times 3 \times 3$, stride:(1, 1); padding:(1, 1) <br> conv3: $32 \times 128 \times 1 \times 1$, stride:(1, 1) | $\times 5$ |
| layer3 | conv1: $128 \times 64 \times 1 \times 1$, stride:(1, 1) <br> conv2: $64 \times 64 \times 3 \times 3$, stride:(2, 2); padding:(1, 1) <br> conv3: $64 \times 256 \times 1 \times 1$, stride:(1, 1) <br> downsample.conv: $128 \times 256 \times 1 \times 1$, stride:(2, 2) | $\times 1$ |
| | conv1: $256 \times 64 \times 1 \times 1$, stride:(1, 1) <br> conv2: $64 \times 64 \times 3 \times 3$, stride:(1, 1); padding:(1, 1) <br> conv3: $64 \times 256 \times 1 \times 1$, stride:(1, 1) | $\times 5$ |
| | avgpool | $\times 1$ |
| | fc: $256 \times 10$ | $\times 1$ |

## B.4 Hyperparameters

In table 10, 11, and 12, we summarize the hyperparameter settings for all experiments. If applying our FedGKT framework to a new CNN architecture with different datasets, we suggest tuning all hyper-parameters based on our hyperparameters.

Table 9: Detailed information of the ResNet-109 architecture used in our experiment

| Layer | Parameter & Shape (cin, cout, kernal size) & hyper-parameters | # |
|---|---|---|
| layer1 | conv1: $16 \times 16 \times 1 \times 1$, stride:(1, 1)<br>conv2: $16 \times 16 \times 3 \times 3$, stride:(1, 1); padding:(1, 1)<br>conv3: $16 \times 64 \times 1 \times 1$, stride:(1, 1)<br>downsample.conv: $16 \times 64 \times 1 \times 1$, stride:(1, 1) | $\times 1$ |
| | conv1: $64 \times 16 \times 1 \times 1$, stride:(1,1)<br>conv2: $16 \times 16 \times 3 \times 3$, stride:(1, 1), padding:(1,1)<br>conv3: $16 \times 64 \times 1 \times 1$, stride:(1, 1) | $\times 11$ |
| layer2 | conv1: $64 \times 32 \times 1 \times 1$, stride:(1, 1)<br>conv2: $32 \times 32 \times 3 \times 3$, stride:(2, 2); padding:(1, 1)<br>conv3: $32 \times 128 \times 1 \times 1$, stride:(1, 1)<br>downsample.conv: $64 \times 128 \times 1 \times 1$, stride:(2, 2) | $\times 1$ |
| | conv1: $128 \times 32 \times 1 \times 1$, stride:(1, 1)]<br>conv2: $32 \times 32 \times 3 \times 3$, stride:(1, 1); padding:(1, 1)<br>conv3: $32 \times 128 \times 1 \times 1$, stride:(1, 1) | $\times 11$ |
| layer3 | conv1: $128 \times 64 \times 1 \times 1$, stride:(1, 1)<br>conv2: $64 \times 64 \times 3 \times 3$, stride:(2, 2); padding:(1, 1)<br>conv3: $64 \times 256 \times 1 \times 1$, stride:(1, 1)<br>downsample.conv: $128 \times 256 \times 1 \times 1$, stride:(2, 2) | $\times 1$ |
| | conv1: $256 \times 64 \times 1 \times 1$, stride:(1, 1)<br>conv2: $64 \times 64 \times 3 \times 3$, stride:(1, 1); padding:(1, 1)<br>conv3: $64 \times 256 \times 1 \times 1$, stride:(1, 1) | $\times 11$ |
| | avgpool | $\times 1$ |
| | fc: $256 \times 10$ | $\times 1$ |

Table 10: Hyperparameters used in Experiments on dataset CIFAR-10

| Model | Methods | Hyperparameters | CIFAR-10 | |
|---|---|---|---|---|
| | | | I.I.D. | non-I.I.D. |
| ResNet-56/110 | **FedGKT (ours)** | optimizer<br>batch size<br>edge epochs<br>server epochs<br>communication rounds | Adam, lr=0.001, wd=0.0001<br>256<br>1<br>20<br>200 | SGD, lr=0.005, momentum=0.9<br>256<br>1<br>40<br>200 |
| | FedAvg | optimizer<br>batch size<br>local epochs<br>communication rounds | Adam, lr=0.001, wd=0.0001<br>64<br>20<br>200 | Adam, lr=0.001, wd=0.0001<br>64<br>20<br>200 |
| | Centralized | optimizer<br>batch size<br>epochs | Adam, lr=0.003, wd=0.0001<br>256<br>300 | |
| | Centralized (ResNet-8) | optimizer<br>batch size<br>epochs | Adam, lr=0.003, wd=0.0001<br>256<br>300 | |

Table 11: Hyperparameters used in Experiments on dataset CIFAR-100

| Model | Methods | Hyperparameters | CIFAR-100 | |
|---|---|---|---|---|
| | | | I.I.D. | non-I.I.D. |
| ResNet-56/110 | **FedGKT (ours)** | optimizer<br>batch size<br>edge epochs<br>server epochs<br>communication rounds | Adam, lr=0.001, wd=0.0001<br>256<br>1<br>20<br>200 | SGD, lr=0.005, momentum=0.9<br>256<br>1<br>40<br>200 |
| | FedAvg | optimizer<br>batch size<br>local epochs<br>communication rounds | Adam, lr=0.001, wd=0.0001<br>64<br>20<br>200 | Adam, lr=0.001, wd=0.0001<br>64<br>20<br>200 |
| | Centralized | optimizer<br>batch size<br>epochs | Adam, lr=0.003, wd=0.0001<br>256<br>300 | |
| | Centralized (ResNet-8) | optimizer<br>batch size<br>epochs | Adam, lr=0.003, wd=0.0001<br>256<br>300 | |

Table 12: Hyperparameters used in Experiments on dataset CINIC-10

| Model | Methods | Hyperparameters | CINIC-10 | |
|---|---|---|---|---|
| | | | I.I.D. | non-I.I.D. |
| ResNet-56/110 | **FedGKT (ours)** | optimizer<br>batch size<br>edge epochs<br>server epochs<br>communication rounds | Adam, lr=0.001, wd=0.0001<br>256<br>1<br>20<br>200 | SGD, lr=0.005, momentum=0.9<br>256<br>1<br>40<br>200 |
| | FedAvg | optimizer<br>batch size<br>local epochs<br>communication rounds | Adam, lr=0.001, wd=0.0001<br>64<br>20<br>200 | Adam, lr=0.001, wd=0.0001<br>64<br>20<br>200 |
| | Centralized | optimizer<br>batch size<br>epochs | Adam, lr=0.003, wd=0.0001<br>256<br>300 | |
| | Centralized (ResNet-8) | optimizer<br>batch size<br>epochs | Adam, lr=0.003, wd=0.0001<br>256<br>300 | |