[Reviews · NeurIPS 2020]

Review 1

Summary and Contributions: The authors propose a method for collaborative training of large CNNs at the edge, which is not feasible today because of the computational case. Instead, the paper trains small CNNs locally on each device, and then uses knowledge distillation to distill the small CNNs into a larger CNN that runs on the server-side. Importantly, instead of communicating the model weights as in federated learnin, the approach communicates the hidden features to train the server and edge networks.

Strengths: - The contribution is novel, and addresses an important problem within edge device learning. They creatively adapt knowledge distillation to share information in a distributed setting. - Potential applications in health care as well, where high resolution CNNs may need to be learned in a distributed setting - The improvements in efficiency are substantial.

Weaknesses: - The weakness of this method, which should be made more clear, is that by communicating the hidden features, and not the weights, there is now a dependence of bandwidth on the dataset size. This is called out partly in lines 141-143, and lines 172-178, but its unclear if the GKT formulation removes this dataset limitation, since H is still being transferred between the edge and the servers. Any empirical data would have been ideal here, but not required. - Large CNNs can either be deep (e.g. RN-56 and RN-110 addressed here) or have high resolution inputs (e.g. in medical settings). Since both are quite prevalent in industry, the paper's significance could be strengthened by a discussion on how these methods (would or would not) scale to high resolution images. - The privacy preserving properties of this method have not been described [response to rebuttal]. Thanks for taking the time to put the rebuttal together. The authors acknowledged that the privacy preserving is not significantly analyzed in the paper. While the rebuttal provides some response, without data it is not convincing. Especially as the paper compared against other SOTA FL methods, this seems to be a critical weakness. Furthermore, the author's rebuttal states that the feature maps are much smaller than the weights, but use a 32x32 image example. In most real world examples with image sizes in 224x224 (e.g. ImageNet) or 1000x1000 (segmentation datasets), the opposite is true. For example, a typical 3x3 conv layer would have H=128, W = 128, and C=K=64. Then, the feature map is CHW ~ 1M elements, whereas weights are 3x3xCxK ~ 37K elements. This indicates the the method's communication characteristics measured in this paper do not scale to real-world CV environments. However, understand that in some cases, the trade-off may be worthwhile between on-device compute and communication. For these reasons, and having read through the other rebuttals, I have downgraded my score.

Correctness: Yes

Clarity: Yes

Relation to Prior Work: Yes, although the prior art baselines (e.g. FL benchmark) are not directly comparable due to the size of CNNs. Yet, the paper does benchmark on small CNNs (e.g. Table 5) with just a few CNN layers, so its unclear why the FL benchmark could not have been used here.

Reproducibility: Yes

Additional Feedback:


Review 2

Summary and Contributions: This paper presents Group Knowledge Transfer (GKT), a novel federated learning algorithm that takes the advantage of knowledge distillation on both clients-to-server and server-to-clients sides to both improve the global model quality and communication efficiency in federated learning applications. Moreover, in GKT the clients only need to train compact model architectures, which improves the training efficiency especially under the scenarios where participating clients are mobile devices. Extensive experimental results are provided to demonstrate the effectiveness of GKT in federated learning scenarios.

Strengths: The paper is well-written. The research area of improving the communication efficiency and effectiveness of existing federated learning algorithms is promising. The advantage of allowing clients to train smaller local models in GKT is convincing, which can improve the on-device training efficiency when participating clients are mobile devices. I commend the authors for providing extensive experimental results.

Weaknesses: The concerns on this paper are summarized below, I will be happy to improve my evaluating score if the concerns are addressed: (1) In the FedAvg [1] algorithm, not all clients will be sampled to participate in a particular federated learning round. However, in GKT, it seems (from Algorithm 1) all available clients will participate in every federated learning round. Will it be possible for GKT to support subsampling a batch of clients for each federated learning round? The subsampling approach will also save communication since less number of clients are allowed to communicate with the data center. (2) From the experimental results, it seems GKT works well for convolutional neural networks. But the knowledge distillation framework seems to be easy to extend to language models e.g. LSTMs and the Transformer architectures [2]. It would be helpful to show the effectiveness of GKT under several NLP tasks e.g. language translation and sentiment analysis. (3) FedMD [3] also uses knowledge distillation in federated learning applications. It would be useful to compare GKT with FedMD to understand the effectiveness of GKT better. [1] https://arxiv.org/pdf/1602.05629.pdf [2] https://arxiv.org/abs/1706.03762 [3] https://arxiv.org/abs/1910.03581 ----------------------------------------------------------------------------- update after the author response: I appreciate the authors for providing the response, which addresses part of my concerns on the scalability of the proposed method and its compatibility with the subsampling approach people usually take in FL. The proposed method is novel to me although the privacy issue remains open for further discussion. At this stage, I tend to remain my overall evaluation score unchanged. -----------------------------------------------------------------------------

Correctness: The main claims and methods proposed in this paper seem to be correct. The empirical methodology also seems to be correct.

Clarity: The paper is well-written. The motivation for this work is promising. The experimental study is extensive and thorough.

Relation to Prior Work: This paper proposed a double side knowledge distillation algorithm, which is designed in specific for federated learning applications. Most of the previous work requires the model architecture on the clients and on the data center to be the same. GKT instead allows the local clients to train smaller and compact models, which can improve the local training efficiency especially under the application scenario where local clients are mobile devices.

Reproducibility: Yes

Additional Feedback:


Review 3

Summary and Contributions: The paper describes an alternating minimization approach to training large CNNs on decentralized data (multiple edge devices each with a local dataset). The approach offers communication and local computation advantages over previous approaches such as split learning and the FedAvg algorithm for federated learning.

Strengths: This is an interesting and relatively novel approach. Experiments, including ablation studies, do a good job of demonstrating the value of the new approach. Hyperparameters are well documented. The communication and computation savings demonstrated are large.

Weaknesses: The pseudocode is critical to understanding the proposed approach precisely, however notational inconsistencies make it considerably harder to understand. See comments under “clarity” below. In order for this work to be well motivated, more details need to be provided indicating the real-world scenarios for which it might be helpful, and how constraints or characteristics of those settings are addressed by the algorithm. For example, the description seems to imply all clients participate in every round, which would rule out the application to cross-device FL setting (See [A] Table 1). Similarly, it is worth clarifying whether clients need to maintain state across rounds, which is typically also not possible in cross-device settings. How do the algorithms perform when on each round devices are sampled from a very large set? (See comments below about optimizer state in particular). Some of the discussion in [A] Sec 1.2 could also be used to improve the motivation. -------------------- Update based on the author response: Thank you for addressing the above point. Please make sure this point is also addressed in the revised version, particularly being explicit about client state and what assumptions need to hold for this approach to apply. In particular, I'm not sure that the "pre-defined client selection strategy" sketched is practical from an infrastructure perspective. See e.g. https://arxiv.org/abs/1902.01046 which describes the many constraints that a production system might face which could limit the ability of groups of clients to participate repeatedly. -------------------- Privacy is one of the primary motivations for federated learning and other collaborative training techniques, and unfortunately that is not addressed. A key principle in FL specifically is using aggregates of user messages, which allows composability with other privacy-enhancing technologies including secure aggregation (the server only sees the sum of updates, not any individual device’s update) and differential privacy. See the definition of FL in [A], as well as Sec 4. Unfortunately, the features maps H^k and labels Y^k might reveal significant private information about client $k$ (in fact, the whole point seems to be that H^k preserves as much semantic information about the examples as possible). It would be preferable to test the algorithm on standard benchmark datasets, for example [B] proposes a natural non-IID partitioning of CIFAR-10 and gives strong baselines for FL on this dataset. By using such a standard federated dataset, comparison to the baselines in this work would be possible. It is implied that there is significant communication savings relative to FedAvg, but this comparison has not been done explicitly (either via a table showing communication cost per round for each algorithm, or by using total bytes communicated as the x-axis in e.g. Fig 3). It is important that the ablation study is included, though currently it isn’t clear if the ablation study was on the IID or non-IID versions of the datasets; please report results for both. These experiments also highlight two points that deserve further attention. First, “diverges” for the problem without the KT loss functions is unconvincing (I assume this is equivalent to solving Eqs (4) - (5) and then (2) and (3), but this should be clarified). I would expect with an appropriately chosen optimizer and learning rate you should get some results from this approach. Second, the differences between “Only S -> E” and “Both” are small, and it’s not clear if they are significant (were the results compared across multiple different randomized initializations, for example?). The takeaway seems to be that S -> E is what matters, and the early parts of the paper should be written with this in mind (For example, when the KT losses are introduced, it would be good to mention this result already). [A] Advances and Open Problems in Federated Learning. https://arxiv.org/abs/1912.04977 [B] Adaptive Federated Optimization. https://arxiv.org/abs/2003.00295.

Correctness: I believe the claims are correct and the empirical methodology is sound.

Clarity: The paper is generally well written (a few suggestions for improvements below). However, Algorithm 1 (line 171) should be improved, which would make the approach in the paper much easier to understand. There is a confusing change in notation; the rest of the paper uses H^{(k)}_i for the client feature maps, but they appear to be denoted X^k_e here, which is quite confusing. ClientUpdate (implicitly) depends on the values Z_s[k] broadcast from the server. This should be made more explicit; first when ClientLocalTraining first executes (Line 4), these values have not yet even been broadcast (line 14). It would perhaps be better to include Z_s[k] explicitly as an argument in the call to ClientLocalTraining, with a comment that the arguments are broadcast from the server. Second, technically Z_s[k] could change every time W_e^(k) changes, so it wasn’t obvious that these are fixed in ClientLocalTraining. Using the broadcast values explicitly in the call to \grad \ell_c would help with this. Lines 21, 25, and 26 in the pseudocode mention \alpha, which is not defined elsewhere. They also use a lower-case w rather than W as in the rest of the paper. Please also be consistent about f^k vs f^(k) to make it clear to the reader these are the same things. Line 29: In addition to X^k_e (which I’m assuming is H^{(k)}), the server also needs Y^k, which should be made explicit. This should also be clarified on line 182 in the body of the paper. Some of the additional comments below also address clarity issues.

Relation to Prior Work: Yes.

Reproducibility: Yes

Additional Feedback: Local comments: Line 12: “maintaining model accuracy comparable to FL” ==> FL is a setting rather than a specific algorithm, so “comparable to FedAvg for FL” would be more precise. Eq (4) and (5): It is worth emphasizing that this optimization can be solved independently on each client, and there is no dependence on any server-side variables, so Eq. (2 - 5) could be solved using a single round of communication. Lines 143 - 146: It is unclear what “correlated” means here. Eq (6) and (7): Showing the arguments to the loss functions would improve readability here and in the pseudocode. Eq. (11) defining z^*_s: Line 192 “Tensorflow cannot fully support our GKT algorithm”. I believe it should be straightforward to implement GKT using TensorFlow Federated [C]’s federated_collect and federated_broadcast operators. Line 204 “(e.g., MNIST)”. This confuses the MNIST dataset with the Federated EMNIST dataset provided by LEAF as well as TensorFlow Federated. Fed EMNIST is larger, has a natural user partitioning (the actual writer of the characters) and includes characters from the alphabet as well as digits (more classes). Please clarify. Line 210 “batch normalization” - It is not straightforward to meaningful implement BatchNorm in FedAvg particularly for non-IID data, so this deserves comments. Recently, group normalization has been shown to be effective in the FL setting. [B] Line 229 - 230: The use of alternative optimizers deserves additional detail. Are these applied on the server (as introduced by [B]) or on the client ? If on the client, how is the optimizer state handled across rounds (cleared or maintained)? If it is maintained, that means these results likely don’t apply to the cross-device setting where devices will often only participate once over the course of training (e.g. [A] Table 2,. say 10^6 devices, R=1000, and 1000 devices per round). Figure 3: What does a “communication round” mean for the centralized setting? General, using an x-axis of “rounds” is most meaningful where each algorithm being compared uses a similar amount of bandwidth, and that’s not the case here. Clarify in the legend whether this is the IID or non-IID setting. [C] https://www.tensorflow.org/federated


Review 4

Summary and Contributions: The paper reformulates federated learning (FL) as a group knowledge transfer (GKT) training algorithm. The authors can train small CNNs on clients and periodically transfer their knowledge by knowledge distillation to a large server-side CNN. They evaluate GKT by training ResNet-56 and ResNet-110 on CIFAR-10, CIFAR-100, and CINIC-10 and their non-IID variants. The experimental results show that GKT can achieve comparable or slightly higher accuracy. Update: I have read the authors' rebuttal and the other reviews. I acknowledge the authors for addressing my review. I have increased my score in light of the authors' rebuttal and the reviewer discussion. I hope the authors add their corresponding explanations in the revised manuscript.

Strengths: - The author claims that GKT makes edge training affordable and preserves the data privacy requirements of the FL approach.

Weaknesses: However, there are two main concerns. - In algorithm 1, ClientLocalTraning has to send features and logits to server. [1] shows that even gradients may cause the deep leakage, and we can obtain the private training set from the publicly shared gradients. The features and logits contain much richer information than the gradients. Thus, this work has a serious privacy problem. Then the motivation of this work, applying knowledge distillation to FL, is not convincing. - The authors mentioned GKT requires lower communication cost. Sending features and logits to server may not always save the communication cost because it depends on the sizes of features and logits. [1] Zhu, Ligeng, Zhijian Liu, and Song Han. "Deep leakage from gradients." Advances in Neural Information Processing Systems. 2019.

Correctness: Yes.

Clarity: Yes.

Relation to Prior Work: Yes.

Reproducibility: Yes

Additional Feedback:

[Author Response · NeurIPS 2020]

- **_1. Privacy preserving_**_. R4/R3: the features maps might leak privacy; R1: privacy property has not been described._

We will discuss the privacy concerns in our revision. **1)** our work does not focus on privacy-preserving techniques, but we believe the hidden vector reconstruction attack can be defended by privacy-preserving techniques such as differential privacy (DP) and multi-party computation (MPC). **2)** it is not necessarily straightforward to conclude that a hidden feature map is less safe than the model or gradient. As pointed out by R4, the model or gradient exchange may also leak privacy. At least from the current literature, no paper has investigated the comparison from a theoretical or experimental perspective. **3)** please note that the hidden feature exchange happens at the training phase. This makes the attack harder because what the attacker access is the evolving and untrained feature map rather than a fully trained feature map that represents the raw data. Analyzing the degree of privacy leakage under our framework will be our future works.

- _"2. **Communication Cost**. R4: may not always save the communication cost because it depends on the sizes of features and logits; R3: comparison has not been shown explicitly; R1: a dependence of bandwidth on the dataset size"_

Please note that the sizes of features and logits are fixed by the CNN architecture design. Compared to the entire model weight or gradient, the hidden vector is definitely much smaller (e.g., the hidden vector size of ResNet-110 is around 64KB while the entire gradient/model size is 4.6MB for 32x32 images). The hidden vector for each data point can be transmitted independently, thus GKT has less bandwidth requirement than gradient or model exchange, although the communication cost in total depends on the number of data points. Moreover, our experimental result shown in **Figure 5** even demonstrates that our method has smaller communication costs for the entire training than split learning (a method that also exchanges hidden vectors during training) because of less communication round for convergence. We will explain communication more obviously in our revision.

- _"3. **Scalability with subsampling strategy**. R3: does the proposed method support cross-device setting? Can we use subsampling?How to maintain the optimizer state among clients? R2: subsampling can save communication cost."_

Our method definitely can support the cross-device setting. First, we believe the user selection strategy is still an open problem. It is better to tailor the strategy for different models and optimization methods. The random subsampling method mentioned by R3 and R2 may not fit for our large DNN setting. The random subsampling may cause many users' data to be touched only once. This is not a big issue for shallow NN because shallow NN requires much fewer data to converge than large DNN. However, it is a problem to large DNN because it typically requires all samples to be trained many epochs. Therefore, under GKT framework, it is more reasonable to use a pre-defined client selection strategy. All clients are divided into many groups. We then train group by group to make sure each group is trained multiple rounds (epochs). The optimizer state of each group can be maintained by uploading it to the server of GKT. Once the group ID is changed, the server then synchronizes the optimizer state to the clients in the new group. This training process is essentially the same as our GKT algorithms: from the perspective of optimization, both "viewing 10 users' dataset as an epoch (as done in our experiments)" and "viewing each user' dataset as an epoch and training user by user" can converge. We will demonstrate this in our revision. Besides sampling, client-edge-cloud hierarchical FL is also a potential solution. We can use the edge server in the hierarchical topology to improve load balance. Also, from the perspective of alternating optimization, GKT does not have scalability issues since GKT does not do synchronous aggregation on the server-side: the server can immediately start training once it receives updates from any client.

- _"4: **Ablation Study**. R3: the diverge result is unconvincing; mention the takeaway earlier in the method section; Please provide IID and non-IID experiments for ablation study"_

Good suggestions. The word "diverge" is somewhat misleading. Actually, we find it is hard to tune the parameters if we do not use KD, sometimes it diverges, and sometimes it gets a low accuracy. We will clarify this in our revision. Although we get this takeaway in ResNet, "both" KD loss may be more effective to other models. So it is better to leave it as a hyper-parameter to tune in practice rather simply concluding which direction is useful. Both IID and non-IID experiments will be provided in our revision.

- _"5: **Benchmark datasets**. R3: it is better to use the benchmark dataset by [1] adaptive federated optimization."_

We admit that using benchmark dataset is a good practice for FL research, but please note that our scenario is totally different from [1]: 1) our research focuses on large CNN training, but [1] is shallow neural network research (see the appendix of this paper, only 2 Conv layers are used). Its CV dataset only includes FMNIST and CIFAR-100, which are inadequate to evaluate a large CNN model. In fact, we use datasets that are more difficult than [1], including CIFAR-10, CIFAR-100, and CINIC-10. 2) our method focuses on a new training framework rather than proposing a better FL optimizer. Thus it is unnecessary to align the CIFAR-100 partition the same as [1].

- _"6. additional comments from R3 such as 1) Algorithm 1 notation issues; 2) TensorFlow Federated"_

We sincerely thank R3 for so many useful suggestions. 1) we will modify Algorithm 1 as your suggestion; 2) Compared to TTF, we develop a more flexible send/recv framework (Fig. 6). We will try TTF in our future work.

- _"7. R1: the paper's significance could be strengthened by discussing on the scale to high resolution images"_

Thanks for your suggestion. We will evaluate the efficiency for high resolution images in our future works.

- _"8. R2: comparing with FedMD; extend GKT to language models like LSTM."_

We already discussed different knowledge distillation methods including FedMD in our related works. As for the language model, we extend it as a future work because careful consideration is needed to tailor for the characteristics of LSTM and Transformer (the back-propagation through time in LSTM and attention mechanisms in Transformer).

[Meta-Review · NeurIPS 2020]

The paper provides a new algorithm for Federated Learning with resource constrained edge devices. The algorithm adapts distillation based techniques (which are usually used for model compression from larger model to smaller model) to a kind of a two way knowledge transfer model that aids learning of local small neural networks on the edge devices and a larger global network on the server cloud. Methodologically the paper is novel, useful, and well written. But a few points raised by the reviewers are very pertinent and needs to be discussed in the final version 1. One key advantage & motivation for the model is stated as reduced communication - unfortunately this has not been empirically justified against FedAvg - the method has the potential for less frequent communication compared to FedAvg but this has not been validated empirically -- it would be good to have this information on the experiments reported - exchanging features over parameters is stated as an advantage, but I agree with R1&R3’s concern that this may not be the case in now-standard networks on high resolution images where the per iteration communication scales as #samples x #hidden units (or features), which could be large. I encourage the authors to be upfront and honest about the potential shortcoming and argue the benefit of the small memory footprint in spite of potential higher communication cost. 2. Although the paper is not about privacy, sharing of NN layer activations of data points has potential privacy concerns and I strongly encourage the authors to have a detailed discussion about the (potentially negative) privacy implications (including in the broader impacts section)